# Unveiling the Antioxidant, Cytotoxic, and Anti-Inflammatory Activities and Chemical Compositional Information of an Invasive Plant: *Lycium ferocissimum* Miers

**DOI:** 10.3390/plants13071035

**Published:** 2024-04-06

**Authors:** Müberra Koşar, Gökçe Şeker Karatoprak, Beste Atlı, Selen İlgün, Esra Köngül Şafak, Nesrin Öztinen, Sena Akçakaya Mutlu, Ezgi Ak Sakallı

**Affiliations:** 1Department of Pharmacognosy, Faculty of Pharmacy, Eastern Mediterranean University, North Cyprus, Via Mersin-10, 99628 Famagusta, Türkiye; beste.atli@emu.edu.tr (B.A.); nesrin.oztinen@emu.edu.tr (N.Ö.); ezgi.aksakalli@emu.edu.tr (E.A.S.); 2Department of Pharmacognosy, Faculty of Pharmacy, Erciyes University, 38039 Kayseri, Türkiye; gskaratoprak@erciyes.edu.tr (G.Ş.K.); esrakongul@erciyes.edu.tr (E.K.Ş.); 3Department of Pharmaceutical Botany, Faculty of Pharmacy, Erciyes University, 38039 Kayseri, Türkiye; serturk@erciyes.edu.tr (S.İ.); senaakcakaya@erciyes.edu.tr (S.A.M.)

**Keywords:** *Lycium ferocissimum*, antioxidant, anti-inflammatory, cytotoxicity, chromatography

## Abstract

In this study, the antioxidant (DPPH and ABTS radical-scavenging, ferric-reducing, iron (II)-chelating), anti-inflammatory (LPS-induced Raw 264.7 cell line), and cytotoxic activities (Du145 and A549 cell lines) of raw fruit, ripe fruit and leaves of the *Lycium ferocissimum* species were examined. By using high-pressure liquid chromatography, *p*-OH benzoic acid, caffeic acid, and rutin were detected in the ethanol and water extracts. For the most active raw fruit ethanol extract, the IC_50_ in terms of the DPPH-scavenging activity was 0.57 mg/mL, and the ABTS inhibition percentage was 88.73% at a 3 mg/mL concentration. The raw fruit ethanol extract exhibited significant inhibition of viability in the Du145 cell line in the concentration range of 62.5–1000 µg/mL. Additionally, the extract effectively reduced the LPS-induced inflammation parameters (TNF-α, IFN-γ, PGE 2, and NO) at a concentration of 31.25 µg/mL. The biological activities of *L. ferocissimum*, which have been elucidated for the first time, have yielded promising results.

## 1. Introduction

It is well known that people have relied on plants to treat illnesses and maintain their health since the dawn of time. The therapeutic properties of the organic compounds known as the secondary metabolites of plants make them valuable for use in the treatment of various diseases. These secondary metabolites, which generally exhibit high pharmacological activity, have led to the plant being primarily utilized for its antioxidant, antimicrobial, anti-inflammatory, and wound-healing properties [1]. *Lycium* species have also drawn the interest of many researchers due to their abundant chemical content and their popularity among the public. Members of the genus *Lycium*, which is a member of the Solanaceae family, grow in temperate and subtropical climates [2]. *Lycium barbarum* and *Lycium chinense* have long been used as common traditional Chinese medicines in China. Among the traditional uses of *Lycium* species, their use in the treatment of inflammation draws attention [3]. Phytochemical studies on *Lycium* species revealed that they contain especially polysaccharides, lipids, terpenes, and phenolic compounds [4]. Glycerogalactolipids, phenylpropanoids, coumarins, lignans, flavonoids, amides, alkaloids, anthraquinones, organic acids, terpenoids, sterols, steroids, and their derivatives are among the chemical substances found in members of the genus [5]. In vitro and in vivo studies were carried out to ascertain the anti-inflammatory and antioxidant effects of some *Lycium* species. *L*. *barbarum* is among the most studied species [6,7,8]. Additionally, species such as *Lycium ruthenicum*, *Lycium chinense,* and *Lycium europaeum* have been investigated for their biological activities [9,10,11,12]. *Lycium ferocissimum* Miers., also known as African boxthorn among the species included in the genus, is an invasive weed in environmental and agricultural ecosystems [13].

*Lycium ferocissimum* is native to the Cape region and Orange Free State, South Africa. The perennial shrub *L. ferocissimum* has many branches and can reach heights and widths of up to 5 m. The leaves are glabrous, ovate, obovate to elliptic in shape, simple and whole, with very short petioles; they often cluster at the nodes. The fruit is a smooth round berry that starts out green but ripens to orange–red [14]. It is regarded as a Weed of National Significance in Australia, and stakeholders in agriculture and the environment agree that controlling it is a challenging and expensive task [15]. *L. ferocissimum* is known to negatively affect native animals, displace native flora, deteriorate wildlife habitats, and probably contribute to the deterioration of cultural heritage sites [16]. Although it is not widely distributed worldwide, it has invaded Cyprus, especially coastal and island plant communities, by seed dispersal through birds and small mammals [17]. The fruit of the *L. ferocissimum* species grown in Cyprus are used as food and the aboveground part is used in the treatment of narcotic poisoning [18], but no research has been conducted to determine the species’ biological activities.

Currently used drugs have many side effects, such as gastrointestinal ulceration and bleeding, kidney damage, hypertension, hyperglycemia, and many more. Besides these side effects, the major disadvantage of currently prescribed drugs is their toxicity and the recurrence of symptoms after the discontinuation of treatment. Efforts are currently underway to screen and develop new compounds for their biological activities, as well as to discover new active secondary metabolites and herbal drugs from medicinal plants. Unlike modern allopathic medicines, which have single active ingredients that target a specific pathway, herbal medicines work based on a collaborative approach. Numerous chemicals found in plants work in concert to affect specific components of the intricate biological system [19]. Species of the genus *Lycium* commonly serve as sources of food and medicine, but despite their popularity, there appears to be insufficient scientific evidence available regarding the use of *Lycium*. For this reason, within the scope of this study, *L. ferocissimum*, which is found in the flora of Cyprus and does not have a detailed activity and phenolic composition study, was evaluated for the first time in terms of its antioxidant, cytotoxic, and anti-inflammatory activities. The activities of ethanol and water extracts prepared from raw fruit, ripe fruit, and leaves of the plant were examined in detail, and their chemical compositions were elucidated by high-performance liquid chromatography (HPLC). Thus, it is believed that the results obtained in line with the aforementioned objectives will contribute to the literature as preliminary studies necessary for obtaining the active drug substance.

## 2. Results and Discussion

### 2.1. Chemical Analysis

The procedure described in the experimental part was used to spectrophotometrically determine the extracts’ total phenolic component levels. The results are presented in Table 1. In terms of the total phenol amounts, the extracts were found to be in the following order: URFEtOH > URFW > LEtOH > LW > RFEtOH > RFW. It is thought that the decrease in the phenolic substance content in ripe fruit may be due to its transformation into other compounds during the biosynthesis process. In a study investigating Lycium shawii fruit extract, the total phenolic and flavonoid contents were found to vary between 100 and 377 mg_GAE_/g_extract_ and between 3.3 and 110.6 mg_quercetin_/g_extract_ [20]. In another study, the dry fruit extracts of *L. barbarum* from cultivation areas in China and Nepal were examined for their total phenolic content. The highest phenolic content of 14.13 mg_GAE_/g_extract_ was found in the Nepalese sample [18]. When comparing our results with this study, we can conclude that *L. ferocissimum* from Cyprus has a higher phenolic content than cultivated *Lycium* species from China and Nepal. 

Different types of chemical constituents have been reported in phytochemical studies of *Lycium* species, including alkaloids, cyclopeptides, lignans, anthraquinones, coumarins, flavonoids, terpenoids, sterols, and other compounds [4]. In our analysis, we qualitatively determined and quantified the amounts of *p*-OH-benzoic acid, caffeic acid, and rutin in the extracts. *p*-OH-benzoic acid was analyzed at 280 nm, caffeic acid at 320 nm, and rutin at 360 nm. The amounts of these identified compounds in the extracts were determined using the calibration curves obtained from standard substances and the results are presented in Table 1. Chromatograms of the raw fruit ethanol extract at 280 nm, 320 nm, and 360 nm are shown in Figure 1.

Among the compounds, the highest amount of rutin was discovered in the leaf water extract, with a value of 0.21745 ± 0.26 (% ± SD). Although HPLC analysis of *L. ferocissimum* is not available in the literature, the results obtained in this study were found to be compatible with the *L. barbarum* species. Luteolin, apigenin, and acacetin derivatives have been identified in *L. barbarum* fruit [3]. In a separate study, it was indicated that *L. barbarum* fruit contained quercetin-diglycoside, rutin, and kaempferol-O-rutinoside. The phenolic acid fraction also contained chlorogenic acid, caffeoylquinic acid, caffeic acid, and p-coumaric acid [6]. The compounds obtained from the ethyl acetate fraction of *L. chinense* included acetin, apigenin, p-coumaric acid, kaempferide, isoscopeolin, caffeic acid, luteolin, kaempferol, vanillic acid, gentisic acid, and linarine [9]. It has been determined that the 80% ethanol extract obtained from the leaves and stems of *L. chinense* contained gallic acid, catechin, gentisic acid, chlorogenic acid, p-hydroxybenzoic acid, vanillic acid, caffeic acid, epicatechin, p-coumaric acid, ferulic acid, sinapic acid, syringic acid, rutin, p-anisic acid, naringin, myricetin, hesperidin, rosmarinic acid, quercitrin, neohesperidin, eriodictyol, diosmin, morin, daidzein, quercetin, naringenin, luteolin, and isorhamnetin [21,22].

### 2.2. Antioxidant Activity

At physiological pH, the extracts’ ability to scavenge the stable nitrogen-centered radical DPPH^●^ was tested, and the results are provided as IC_50_ values (mg/mL). As the IC_50_ value decreased, the activity increased. The IC_50_ values of the *L. ferocissimum* extracts varied between 0.57 and 8.003 mg/mL (Table 2). The most potent extract was the raw fruit ethanol extract, which had an IC_50_ value of 0.57 ± 0.05 mg/mL. The extract with the lowest activity was the water extract of ripe fruit, with an IC_50_ value of 8.003 ± 1.19 mg/mL. In the study conducted by Faidi et al. (2016) on *L. ferocissimum* fruit, carotenoid-rich acetone, ethyl acetate, and hexane extracts were prepared. The IC_50_ values for the DPPH radical-scavenging were reported as 0.65–2.15 mg/mL [23]. A direct comparison cannot be made since the study used different solvents for the extraction. According to Zhang et al. (2013), the antioxidant capacity of *L. barbarum* fruit varied with different concentrations (10 mg/mL, 20 mg/mL, 40 mg/mL, and 50 mg/mL), giving inhibitory percentages of 70.58%, 65.21%, 59.94%, and 52.99%, respectively [24]. These findings are similar to our analysis. The low phenolic content of *Lycium* species is thought to be the reason why they do not exhibit strong antioxidant activity.

In the ABTS^+^ radical-scavenging activity experiment, raw fruit ethanol extract was discovered to be more effective than the other extracts (88.73 ± 5.17%). Despite the fact that all the extracts scavenged the ABTS radical in a concentration-dependent way, the data are only provided at the common concentration point of 3 mg/mL. Other extracts that were discovered to be active were identified as ethanol and water extracts derived from the leaves. In previous research, the IC_50_ value of *L. barbarum* polysaccharide extract for scavenging ABTS^+●^ was determined to be 47.158 ± 6.231 µg/mL. Since polysaccharide extract was not evaluated in this study, it is not possible to compare the results [24]. Due to the lack of previous research studies examining the ABTS radical-scavenging activity of *L. ferocissimum*, the results obtained in this study will help to fill this gap. The chelating effects on the ferrous ions of the extracts were studied in the concentration range of 1–20 mg/mL. The leaf water and raw fruit ethanol extracts were found to be the most active extracts, with IC_50_ values of 2.05 and 3.09 mg/mL. No extract showed as much activity as Na_2_EDTA, which was studied in the 5–100 µg/mL concentration range as a standard. Table 2 displays the results. The reducing power of the extracts from iron (III) to iron (II) was calculated as equivalent to ascorbic acid (AscAE). Among the extracts, the raw fruit ethanol extract was found to have the same significance (*p* > 0.05) as BHT (2.27 ± 0.01 AscAE [mmol/g]), with a value of 1.85 ± 0.004 AscAE [mmol/g]. The other extracts showed similar activity in the range of 0.75–0.82 AscAE [mmol/g]. When comparing the iron (II)-chelating activity of the extracts with the results concerning the free radical-scavenging activities, there is a partial correlation. Secondary metabolites such as phenolic ring-bearing phenolic acids and flavonoids are characterized by their strong antioxidant activities. The fact that the extracts’ phenolic and flavonoid contents are low may be attributed to their lack of strong metal reduction capacity and metal-chelating properties.

### 2.3. Cytotoxic Activity

In the Du145 cell line, the raw fruit 80% ethanol extract exhibited the strongest cytotoxic activity among the other extracts studied. Viability was determined to be significant between the 62.5 and 1000 µg/mL concentrations. The leaf 80% ethanol extract demonstrated a significant decrease in viability compared to the control at concentrations of 500 and 1000 µg/mL (67.75% and 55.22%). The ripe fruit ethanol extract, ripe fruit water extract, raw fruit water extract, and leaf water extract did not lead to a substantial decrease (*p* > 0.0.5) in viability compared to the control at each of the investigated concentrations (Table 3). 

In the study conducted by Ran et al. (2020), the effect of *L. chinense* pollen on Du145 prostate cancer cells was evaluated. The tumor suppression level of tumors in mice administered 100 mg/kg, 200 mg/kg, and 400 mg/kg of *L. chinense* pollen was found to be 40.16%, 57.98%, and 62.31%, respectively [25]. Mottaghipisheh et al. (2022) discovered that at doses of 100–400 µg/mL, the anthocyanin monomer petunidin isolated from *L. ruthenicum* reduced cell growth and triggered apoptosis in the S phase of the Du145 cell line [26]. In another study, it was found that *L. barbarum* polysaccharides substantially and dose-dependently suppressed the proliferation of Du145 cells. Inhibition rates of 23.5%, 39.7%, 57.5%, 82.5%, and 92.5%, were observed at concentrations of 100, 200, 400, 800, and 1000 µg/mL, respectively, in Du145 cells on the 5th day [27]. These findings indicate that extracts or polysaccharides from *Lycium* species do not have high cytotoxic action on cancer cells.

The viability of the A549 cell line was significantly reduced by the raw fruit water extract at concentrations of 500 and 1000 µg/mL, the leaf ethanol extract at a concentration of 1000 µg/mL, and the ripe fruit ethanol extract at a concentration of 1000 µg/mL in comparison to the control. However, the other extracts did not cause a remarkable decrease in viability in comparison to the control at each of the investigated concentrations. In accordance with the research conducted by Ghali et al. (2015), in which the cytotoxic effects of extracts prepared from *L. europaeum* fruits on A549 and PC12 cells were examined using the MTT method, a substantial loss of cell viability was observed in cancer cells at concentrations of 25, 50, 75, and 100 µg/mL [28]. When the data from all of these research studies are compared, it is seen that the antitumor activities of *L. chinense* pollen, *L. barbarum* polysaccharides, and *L. europaeum* fruit extract at similar concentrations were significantly greater than the cytotoxic potential of *L. ferocissimum* extracts.

### 2.4. Determination of Anti-Inflammatory Effect

In cytotoxicity studies conducted with Raw 264.7 cells, the non-toxic doses were determined as 15.625 and 31.25 µg/mL (Figure 2). 

By measuring the pro-inflammatory cytokines generated by LPS-induced Raw 264.7 murine macrophage cells, the effect of the *L. ferocissimum* extracts on the immune system was investigated. There was an increase in PGE2 in the LPS-given group in the experimental model. Among the extracts, the two extracts that significantly reduced the amount of PGE2 were determined to be the ethanol extract of the leaves and the ethanol extract of the raw fruit. The amounts of PGE2 were found to be 1952.27 ± 15.80 pg/mL and 1884.64 ± 9.45 pg/mL, respectively. TNF-α was upregulated in the well where LPS was administered, indicating that inflammation had taken place, but extracts were able to lower the TNF-α in all the wells relative to the LPS group. The most active extract was found to be the raw fruit ethanol extract, with an amount of TNF-α of 1856 ± 19.40 pg/mL at a concentration of 31.25 µg/mL. The raw fruit ethanol and leaf ethanol extracts were shown to be the most potent extracts in the measurement of IFN-γ and NO. While the amount of IFN-γ in the 15.625 µg/mL group to which the raw fruit ethanol extract was applied was 105.19 ± 9.5 pg/mL (*p* < 0.05), this amount was 96.72 ± 2.83 pg/mL (*p* < 0.01) in the 31.25 µg/mL group. The raw fruit ethanol and leaf ethanol extracts exerted remarkable activity (*p* < 0.01) by reducing the amount of NO in the LPS groups, which was 75.17 ± 0.01 µM, to levels of 31.75 ± 7.17 µM and 38.12 ± 5.18 µM. The results are presented in Table 4. The anti-inflammatory potential of *L. ferocissimum* has been clarified for the first time, but *L. barbarum* has been the subject of numerous anti-inflammatory activity investigations [29]. *L. barbarum* fruit extracts have been shown to inhibit LPS-induced inflammation in rats by reducing the TNF-α and IL-6 levels [30]. In another study, it was stated that in the COX-2 peroxidase assay, a greater inhibition capacity was found with *L. ruthenicum* fruit extracts compared to *L. barbarum* extract. This was attributed to the strong capacity of *L. ruthenicum* extract to suppress the COX-2 gene [31]. Studies have revealed the basis for utilizing *Lycium* species as anti-inflammatory agents in traditional treatments, and *L. ferocissimum* has been discovered to have similar anti-inflammatory properties to other species.

## 3. Materials and Methods

### 3.1. Plant Material and Extraction Process

Raw fruit (unripe, immature), ripe fruit, and leaf parts of the *Lycium ferocissimum* plant were collected from the Salamis region of Cyprus (on 17 April 2021). The voucher samples of the plant were stored at the Eastern Mediterranean University, Faculty of Pharmacy. The dried materials were pulverized and extracted with 80% ethanol in a shaker at room temperature. Additionally, 5% decoctions of all the parts were prepared with water. The obtained extracts were concentrated (using an evaporator) and stored at −18 °C until analysis. The extracts were coded as follows: URFEtOH: Raw fruit 80% ethanol extract, URFW: Raw fruit water extract (decoction), RFEtOH: Ripe fruit 80% ethanol extract, RFW: Ripe fruit water extract (decoction), LEtOH: Leaf 80% ethanol extract, LW: Leaf water extract (decoction).

### 3.2. Chemical Analysis

#### 3.2.1. Total Phenolic Content

The total amount of phenolic substances contained in the extracts was calculated as equivalent to gallic acid using the Folin–Ciocalteu method. Here, 100 µL of sample solution and 500 µL of Folin–Ciocalteu reagent were added to a tube containing 6 mL of distilled water. Then, 1.5 mL of 20% aqueous Na_2_CO_3_ (Sodium Carbonate) was added 1 min later and completed with 10 mL of water. After incubation at 25 °C for 2 h, the absorbance at 760 nm was measured and compared with the gallic acid calibration curve [32].

#### 3.2.2. HPLC Analysis

An Agilent 1260 Liquid Chromatography system with a Photodiode Array (PDA) detector (Waldbronn, Germany) was used for the chromatographic analysis. With a flow rate of 1 mL/min, separations were performed using an Inertsil-C18 reverse-phase analytical column (Varian, Torrance, CA, USA). For the injection, extracts were made up at a concentration of 2 mg/mL. The detection was carried out at the 280, 320 and 360 nm wavelengths. For the separation, three solvent systems were used: solvent A: methanol (10)/water (88)/acetic acid (2) (v/v/v), solvent B: methanol (90)/water (8)/acetic acid (2) (v/v/v), and solvent C: methanol. The retention time and UV spectra of the *p*-OH benzoic acid, caffeic acid, and rutin peaks were determined by comparing them with their standards. Injections of the standard and sample solutions were carried out in triplicate.

### 3.3. Antioxidant Activity

#### 3.3.1. DPPH^●^-Scavenging Activity

The DPPH^●^-scavenging capacity of the extracts was measured using the method developed by Gyamfi et al. (1999). Here, 50 µL of extract samples prepared at various concentrations was combined with 450 µL of Tris-HCl buffer and 1.0 mL of 0.1 mM DPPH^●^ in MeOH. The sample absorbance was measured at 517 nm after 30 min of incubation in darkness and at optimal temperature (UV-1800, Shimadzu Corporation, Kyoto, Japan). Equation (1) was used to obtain the percentage of inhibition. Sigma Plot (version 7.0, 2001) was used to compute the IC50 values using the percent inhibitions. As a control, butylated hydroxytoluene (BHT) was utilized [33].
% inhibition = [(Abscontrol − Abssample)/Abscontrol] × 100 (1)

#### 3.3.2. ABTS^+●^-Scavenging Activity

The ABTS radical (7 mM) was produced by letting the ABTS^+●^ aqueous solution and K_2_S_2_O_8_ (potassium persulfate 2.45 mM, final concentration) react for 16 h in the dark, and the absorbance was set to 0.700 (±0.021) at 734 nm. The reaction kinetics were determined at 734 nm for 30 min at 1 min intervals after the extract solution (10 μL) was combined with the radical solution (990 μL). The raw fruit ethanol extract was studied in the concentration range of 0.125–3 mg/mL, while the other extracts were studied in the concentration range of 1–15 mg/mL. Equation (1) was used to obtain the % inhibition. BHT was used as a standard. The mean of three replicate analyses was recorded [34]. 

#### 3.3.3. Ferric-Reducing Activity (FRAP)

Here, 1 mL of extract solution (concentration range 0.01–0.5 mg/mL) was mixed with 2.5 mL of 0.2 M phosphate buffer (pH 6.6) and 2.5 mL of 1% K_3_[Fe(CN)_6_] (potassium hexacyanoferrate) solution and incubated for 30 min (50 °C), and then 2.5 mL of 10% C_2_HCl_3_O_2_ (trichloroacetic acid) was added. After centrifugation for 10 min, 2.5 mL of the supernatant was removed, 2.5 mL of water and 0.5 mL of 1% FeCI_3_ (iron (III) chloride) were added, and mixed, and the absorbance was read at 700 nm. BHT was used as a standard. The results are given as the ascorbic acid equivalent (AscAE) in mmol ascorbic acid/g sample [35].

#### 3.3.4. Iron (II) Chelating

Five minutes after mixing 200 µL of the extract solution with 100 µL of a 2.0 mM aqueous FeCl_2_ (iron (II) chloride) and 900 µL of methanol, the reaction was accelerated with 400 µL of a 5.0 mM ferrozine solution, and the absorbance was read at 562 nm after 10 min. Disodium ethylenediaminetetraacetate dihydrate (Na_2_EDTA) was used as a standard. The results were calculated as IC_50_ values [35].

### 3.4. Cytotoxic Activity

At 37 °C in a 5% CO_2_ environment, the Du145 cell line (ATCC HTB-81TM, human prostate cancer) and the A549 (ATCC CCL-185, human lung cancer) cell line were grown in EMEM and RPMI, respectively. Then, 10% fetal bovine serum (FBS) and 100 μg/L penicillin/streptomycin antibiotic combinations were supplemented to the media of both cell lines.

According to the MTT (3-[4,5-dimethylthiazol-2-yl]-2,5-diphenyltetrazolium bromide) colorimetric method, the proliferated cells were counted and then seeded in a 96-well microplate. Each well contained 10^4^ cells in a volume of 100 µL. After 24 h, the media on the cells attached to the wells were removed. Then, 100 µL of the extracts prepared by dilution in the media from low to high concentrations in the range of 15.625–1000 µg/mL were taken and added to the wells. The plates were kept in a 5% CO_2_ environment at 37 °C for 48 h. Stock MTT solution was obtained in sterile phosphate-buffered saline and added to the plates after dilution. The plates underwent incubation for 3 h. Afterwards, the medium in the wells was discarded and supplemented with 100 µL of dimethyl sulfoxide (DMSO). After 5 min of shaking, the optical densities were read on an ELISA device (Bio-Rad, Hercules, CA, USA) at a wavelength of 540 nm [36].
% Viability = [(Abssample × 100)/Abscontrol](2)

### 3.5. Determination of Anti-Inflammatory Effect

DMEM media (10% FBS, 2 mM L-glutamine, 100 U/mL penicillin/streptomycin) was used to cultivate the murine macrophage Raw 264.7 cell line (ATCC TIB-71, Manassas, VA, USA) and the cells were incubated at 37 °C in a 5% CO_2_ environment. The toxicity of the extracts was assessed at concentrations between 3.9 and 125 µg/mL using the MTT test previously stated.

To determine inflammation, Raw 264.7 cells grown under appropriate conditions were seeded in 6-well plates at 5 × 10^5^ per well. The extracts were dispensed into the wells 3 h before lipopolysaccharide (LPS) application at the 15.625 and 31.25 µg/mL concentrations. Except for the control well, LPS at a dose of 1 µg/mL was administered to all the wells. After 24 h, the supernatant in the plate was collected and centrifuged (10 min, 700× *g*). After applying the extracts, the changes in the TNF-α, IFN-γ, and PGE2 levels were measured using commercial kits (Andygene AD2726Mo, Andygene AD2783Mo, and Andygene AD1630Mo, respectively).

In terms of the nitric oxide measurement, 0.2 g Na_2_B_4_O_7_ was dissolved in 100 mL distilled water, and a standard solution was prepared by adding 0.069 g NaNO_2_. To prepare a standard curve from this prepared solution, dilutions were prepared at concentrations of 5, 10, 20, 40, 80, and 100 µmol/L. Then, 50 µL of Griess reagent was added to 50 µL of supernatants taken from the experimental groups and dilutions of the standard. The resulting mixture was then kept at ambient temperature for 10 min for color formation. After measuring the absorbance at 540 nm, the nitrite concentration of the samples was computed via the standard sodium nitrite calibration curve [35].

### 3.6. Statistical Analysis

Analysis of the variances was carried out utilizing the statistical program SPSS 12 (Inc., Chicago, IL, USA) following the ANOVA procedure. According to the Tukey’s pairwise and Games–Howell comparison tests, significant differences between the means were assessed at the *p* < 0.05 level.

## 4. Conclusions

Based on the findings of this study, the potential anti-inflammatory activity of *L. ferocissimum*, which also has a mild antioxidant and cytotoxic impact, was discovered. It is significant that there has been no previous research conducted on *L. ferocissimum*, and this work aimed to explore its biological activities for the first time. This research will make a valuable contribution to the existing literature. Within the scope of this study, the first important steps were taken to discover compounds with anti-inflammatory effects, and it was once again proven that the secondary metabolites of medicinal plants are one of the most important candidates for becoming drug-active ingredients.

## Figures and Tables

**Figure 1 plants-13-01035-f001:**
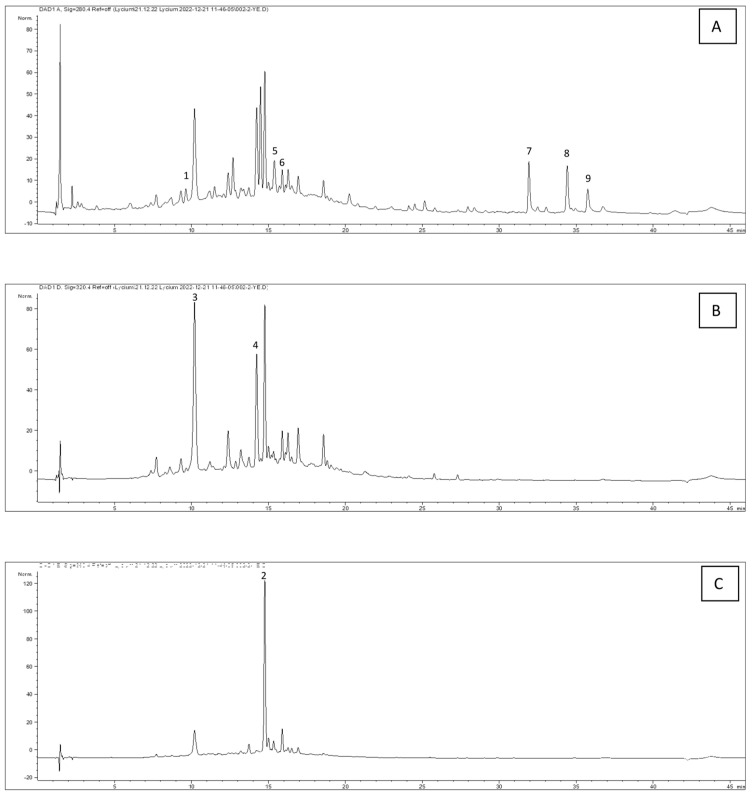
Sample chromatograms of the LEtOH extract at 280 (**A**), 320 (**B**), and 360 (**C**) nm. See the Table 1 for the numbers.

**Figure 2 plants-13-01035-f002:**
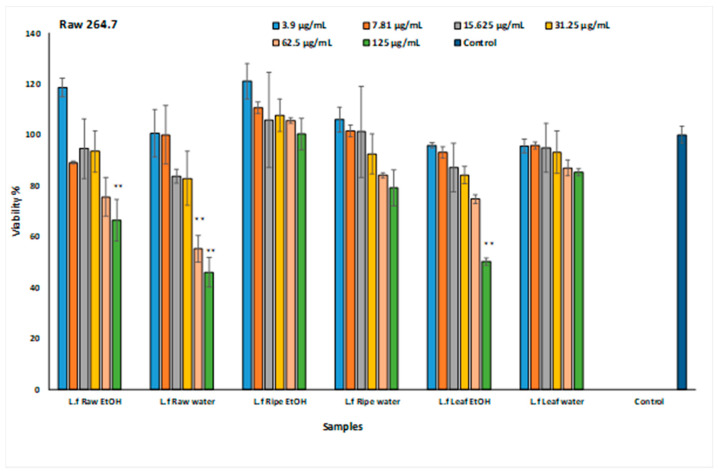
Results of *L. ferocissimum* extracts’ cytotoxic activities on the Raw 264.7 cell line. Values expressed as the mean  ±  sd (n  =  3); statistical evaluations performed with the Games–Howell comparison test. Significant differences are displayed as ** *p* < 0.01. URFEtOH: Raw fruit 80% ethanol extract, URFW: Raw fruit water extract, RFEtOH: Ripe fruit 80% ethanol extract, RFW: Ripe fruit water extract, LEtOH: Leaf 80% ethanol extract, LW: Leaf water extract.

**Table 1 plants-13-01035-t001:** Total phenol content and quantitative determination of *L. ferocissimum* extracts (n = 3).

Extracts *	Total Phenols (mg_GAE_/g_extract_)	*p*-OH Benzoic Acid ***	Rutin ***	Hydroxycinnamic Acid Derivatives ***	Flavonoid Derivatives ***
URFEtOH	49.85 ± 2.13	NI **	NI	15.26 ± 3.23	6.21 ± 0.074
URFW	24.94 ± 0.17	NI	NI	NI	NI
RFEtOH	15.59 ± 3.21	NI	NI	NI	NI
RFW	12.79 ± 1.95	NI	NI	NI	NI
LEtOH	24.14 ± 3.61	0.23 ± 0.15	2.94 ± 0.91	2.23 ± 0.64	9.12 ± 0.33
LW	23.37 ± 4.17	0.10 ± 0.51	5.39 ± 1.12	NI	10.02 ± 3.41
Retention **** Time (Min)		9.64	14.76	6.87, 7.03, 8.99, 10.19, 14.24	15.38, 15.90, 31.91, 34.42, 35.75

* URFEtOH, unripe fruit ethanol extract; URFW, unripe fruit water extract; RFEtOH, ripe fruit ethanol extract; RFW, ripe fruit water extract; LEtOH, leaf ethanol extract; LW, leaf water extract; ** NI, not identified; *** mean _mg/gextract_ ± SD (n = 3); **** Retention time for HPLC peaks (Figure 1).

**Table 2 plants-13-01035-t002:** Antioxidant activity results of *L. ferocissimum* extracts (n = 3).

	DPPH^●^ IC_50_ (mg/mL)	ABTS^+●^ % Inhibition (3 mg/mL)	FRAP mmol Ascorbic Acid/g Sample	Iron (II) Chelating IC_50_ (mg/mL)
URFEtOH	0.57 ± 0.05 ^d^	88.73 ± 5.17 ^a,b^	1.85 ± 0.004 ^b,c^	3.09 ± 0.12 ^c^
URFW	6.27 ± 0.42 ^a^	45.55 ± 4.78 ^c^	0.82 ± 0.005 ^a^	ND *
RFEtOH	6.12 ± 0.90 ^a^	36.37 ± 2.21 ^d^	0.76 ± 0.016 ^a^	12.19 ± 1.21 ^b^
RFW	8.00 ± 1.20 ^b,c^	32.88 ± 1.13 ^d^	0.76 ± 0.002 ^a^	14.71 ± 2.52 ^b^
LEtOH	3.03 ± 0.13 ^b^	54.95 ± 5.62 ^e^	0.77 ± 0.005 ^a^	11.95 ± 1.73 ^b,d^
LW	3.86 ± 0.30 ^b^	58.27 ± 3.25 ^e^	0.76 ± 0.003 ^a^	2.05 ± 0.36 ^c^
BHT Na_2_EDTA	0.008 ± 0.001 ^e^	92.15 ± 2.14 ^a^	2.27 ± 0.01 ^c^	10.44 ± 0.01 µg/mL ^a^

Values given as the mean ± SD are stated within a ±95% confidence interval. * ND, not detected. Significant (*p* < 0.05) differences are shown by different lowercase letters (a–e). URFEtOH: Raw fruit 80% ethanol extract, URFW: Raw fruit water extract, RFEtOH: Ripe fruit 80% ethanol extract, RFW: Ripe fruit water extract, LEtOH: Leaf 80% ethanol extract, LW: Leaf water extract.

**Table 3 plants-13-01035-t003:** Cytotoxic activity results of *L. ferocissimum* extracts on the Du145 and A549 cell lines.

**Du145 Cell Line** **(% Viability)**	**Concentrations (µg/mL)**						
Extracts	15.625	31.25	62.5	125	250	500	1000
URFEtOH	92.88 ± 1.54	87.83 ± 3.99	63.20 ± 3.96 ***	63.95 ± 5.21 ***	58.71 ± 4.87 ***	54.31 ± 0.70 **	25.75 ± 0.32 *
URFW	94.42 ± 0.57	98.99 ± 3.56	95.34 ± 3.08	96.62 ± 4.91	93.60 ± 1.51	93.88 ± 1.98	91.13 ± 0.41
RFEtOH	94.79 ± 2.01	94.22 ± 1.34	87.49 ± 2.84	85.97 ± 0.59	81.71 ± 0.65	77.06 ± 3.98	77.73 ± 2.58
RFW	90.31 ± 2.59	90.31 ± 2.19	89.24 ± 2.07	88.27 ± 2.91	91.28 ± 2.46	89.56 ± 3.67	85.90 ± 3.08
LEtOH	88.23 ± 0.59	88.80 ± 2.01	87.56 ± 1.14	87.56 ± 1.31	81.91 ± 4.06	67.75 ± 2.73 ***	55.22 ± 0.87 **
LW	82.31 ± 5.84	85.38 ± 2.75	81.35 ± 5.34	81.92 ± 9.5	84.62 ± 6.4	73.56 ± 3.82	73.65 ± 4.69
**A549 Cell Line** **(% Viability)**	**Concentrations (µg/mL)**						
Extracts	15.625	31.25	62.5	125	250	500	1000
URFEtOH	95.13 ± 3.81	96.30 ± 1.51	94.45 ± 4.52	91.76 ± 7.32	92.27 ± 2.66	93.28 ± 2.19	81.34 ± 4.10
URFW	90.92 ± 5.87	84.04 ± 10.10	81.53 ± 8.93	77.62 ± 9.86	73.71 ± 7.28	70.58 ± 7.58 ***	63.54 ± 3.39 ***
RFEtOH	99.58 ± 6.60	96.95 ± 1.24	93.76 ± 1.27	90.43 ± 2.77	84.47 ± 4.09	77.67 ± 3.63	59.78 ± 0.63 ***
RFW	91.54 ± 5.59	92.65 ± 2.70	94.45 ± 3.14	98.75 ± 6.03	87.52 ± 2.29	78.92 ± 7.67	72.54 ± 6.03
LEtOH	83.87 ± 4.76	83.36 ± 1.04	83.87 ± 5.55	81.34 ± 6.58	79.50 ± 6.71	76.47 ± 7.34	66.05 ± 3.15 ***
LW	92.96 ± 1.24	90.77 ± 1.77	88.89 ± 0.27	86.23 ± 1.64	84.04 ± 3.38	78.72 ± 0.97	73.08 ± 3.39

Values expressed as the mean ± sd (*n* = 3); statistical evaluations performed with the Games–Howell comparison test. Significant differences are presented as * *p* < 0.001 and ** *p* < 0.01 *** *p* < 0.05. URFEtOH: Raw fruit 80% ethanol extract, URFW: Raw fruit water extract, RFEtOH: Ripe fruit 80% ethanol extract, RFW: Ripe fruit water extract, LEtOH: Leaf 80% ethanol extract, LW: Leaf water extract.

**Table 4 plants-13-01035-t004:** Effects of *L. ferocissimum* extracts on the levels of TNF-α, IFN-γ, PGE 2 levels, and NO.

	TNF-α (pg/mL)		PGE_2_ (pg/mL)		IFNƔ (pg/mL)		NO (µM)	
Extract	15.625 µg/mL	31.25 µg/mL	15.625 µg/mL	31.25 µg/mL	15.625 µg/mL	31.25 µg/mL	15.625 µg/mL	31.25 µg/mL
URFEtOH	2018.65 ± 13.61 ***	1856 ± 19.40 **	2056.79 ± 9.85 ***	1952.27 ± 15.80 **	105.19 ± 9.5 ***	96.72 ± 2.83 **	38.17 ± 4.12 **	31.75 ± 7.17 **
URFW	2700.12 ± 10.73	2548.78 ± 19.43 ***	2234.55 ± 15.42	2202.94 ± 18.77 ***	114.03 ± 8.10 ***	99.42 ± 12.56 **	50.19 ± 2.17 ***	45.16 ± 2.79 ***
RFEtOH	2801.25 ± 15.71	2710.56 ± 20.98	2125.58 ± 11.43 ***	2107.64 ± 19.43 ***	109.68 ± 7.14 ***	100.15 ± 6.48 **	52.43 ± 6.42 ***	47.78 ± 3.75 ***
RFW	2805.99 ± 12.13	2698.49 ± 13.62	2392 ± 7.38	2305.45 ± 19.13	121.23 ± 7.78	109.13 ± 9.15 ***	60.76 ± 1.58	54.15 ± 5.41 ***
LEtOH	2144.53 ± 5.66 ***	1978.14 ± 17.14 **	1974.16 ± 10.34 **	1884.64 ± 9.45 **	107.81 ± 13.04 ***	99.15 ± 5.00 **	48.44 ± 3.14 ***	38.12 ± 5.18 **
LW	2244.48 ± 28.25 ***	2151.45 ± 5.89 ***	2249.83 ± 14.41 ***	2056.87 ± 11.45 ***	113.05 ± 9.52 ***	101.45 ± 7.34 ***	57.79 ± 6.19 ***	51.53 ± 8.46 ***
Control		1053.62 ± 9.18 *		1472.46 ± 11.89 *		62.26 ± 3.24 *		7.36 ± 0.01 *
LPS group		2852.82 ± 7.94		2518.54 ± 7.49		140.78 ± 5.25		75.17 ± 0.01

Values expressed as the mean ± sd (*n* = 3); statistical evaluations with the Games–Howell comparison test. Significant differences are displayed in the form of * *p* < 0.001 and ** *p* < 0.01 *** *p* < 0.05. URFEtOH: Raw fruit 80% ethanol extract, URFW: Raw fruit water extract, RFEtOH: Ripe fruit 80% ethanol extract, RFW: Ripe fruit water extract, LEtOH: Leaf 80% ethanol extract, LW: Leaf water extract.

## Data Availability

The data presented in this study are available in the article and no supplementary material has been added.

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
