# Peer review of "Unveiling the Antioxidant, Cytotoxic, and Anti-Inflammatory Activities and Chemical Compositional Information of an Invasive Plant: Lycium ferocissimum Miers"

_plants, 2024, doi:10.3390/plants13071035_

Round 1

Reviewer 1 Report

Comments and Suggestions for Authors

The biological activities of organic extracts and water extracts from raw fruit, ripe fruit, and leaves of Lycium ferocissimum Miers. were determined in this paper. In my opinion this is very well prepared manuscript. There are no data about the biological activities of L. ferocissimum. For that the studies are innovative. However, the description in the results and discussion section is still a little simple, the processing of data is weak, and some modifications need to be made.

Author Response

Dear reviewer,

File is added. 

Reviewer 2 Report

Comments and Suggestions for Authors

Unveiling the Phytochemical Profile, Antioxidant, Cytotoxic, and Anti-Inflammatory Activities of an Invasive Plant: Lycium ferocissimum Miers.

The title is misleading – the phytochemical profile was not ‘unveiled’. The authors focused only on three very popular compounds widely distributed in the plant kingdom: p-hydroxybenzoic acid, caffeic acid, and rutin. Only three out of six tested extracts contained the selected compounds, and the analysis does not provide the quantitative data but only the percentage content.   

The Introduction section does not provide sufficient information about the species Lycium ferocissimum, and the scientific problem is not clearly presented. The statement about the invasive nature of the plant is only in the title. An attempt at a short review of the Lycium genus is too general and inconsistent. The background of the study is lacking. 

The Results and Discussion section needs major revision in terms of presented data. In this section 'results' mostly replicate the tables, while the 'discussion' is limited to listing already published activities from the genus Lycium. 

Statement: 'In terms of total phenol amounts, the extracts were found to be in the following order: L.f Raw EtOH > L.f Raw water > L.f Leaf EtOH > L.f Leaf water > L.f. Ripe EtOH > L.f Ripe water' provides no information. The authors do not discuss what may stem from this kind of consideration. The chemical analysis section is more informative about L. barbarum than about L. ferocissimum.

The chromatograms enclosed are low quality, and none of the compounds selected for analysis are marked on them. Also, UV spectra or retention time are not marked. According to Table 1. L.f. Raw EtOH contains only one out of three selected compounds, so why chromatograms of this extract are presented?

Antioxidant ability - section needs rewriting. The authors mix the results obtained with the discussion in a confusing way. F. ex. in the sentence: ‘In the study conducted on L. ferocissimum fruits, carotenoid-rich acetone, ethyl acetate, and hexane extracts were prepared.’ should be noted that the Authors refer to a different paper. Sentence lines 141-143 is ununderstandable.

Cytotoxic activity. 

Line 177-179: this all stands in Table 3. 

Line 183: ‘compared to the control’ determines the control. Why the control is not enclosed in Table 3?

Materials and Methods - this section needs to be rewritten as it does not fully allow for study repetition. Avoid using the full chemical name in the text (f. ex. ‘carbon dioxide’, ‘sodium nitrate’, etc.). Check this section for equipment relevance – add details concerning the model and producer (f. ex. HPLC column, spectrophotometer). In the section 3.2.2. selected PDA wavelengths are 240, 320, and 360 nm, while chromatograms are captioned 280, 320, and 360 nm. The Material and Method section does not fully correspond to the information in the text (due to Material and Method IC50 was established only for iron chelating omitting the DPPH). 

Some examples:

3.1. Plant Material and Extraction Process 

Specify the difference between “raw’ and “ripe’ fruits. “Raw’ is immature, green form?

The plant organ from which decoction was prepared was not determined also the relevant code for ‘5% decoction’ is lacking. Also ‘decoction’ is mentioned only in the Materials and Methods section.

Line 326 delete ‘waiting time’

3.4. ‘Cytotoxic Activity’ needs rearrangements and checking for English language correction.

( ‘Each well contained 104 cells in a volume of 100 μL.’ – or 104 cells?; carbon dioxide – replace with CO2 but specify the concentration, correct phrases like: ‘the medium in the wells was emptied’, etc.)

3.5. ‘Determination of Anti-inflammatory Effect’ needs to be rewritten and checked for English language correction. For TNF-α, IFN-γ, and PGE2 commercial kits were used according to provided protocol. In the case of Griess test spectrophotometric analysis was performed referring to authentic standard. The procedures are unclear, and the order in which they were carried out is confusing and does not allow to repeat the study. 

Statistical analysis does not determine the number of repetitions of each test/sample.

Comments on the Quality of English Language

The quality of the English language is acceptable, although all sections need rewriting.

Author Response

Dear reviewer,

File is added. 

Reviewer 3 Report

Comments and Suggestions for Authors

Accept in present form

Author Response

Dear reviewer,

File is added. 

Reviewer 4 Report

Comments and Suggestions for Authors

Unveiling the Phytochemical Profile, Antioxidant, Cytotoxic, and Anti-Inflammatory Activities of an Invasive Plant: Lycium ferocissimum Miers.

The scientific study focuses on the biological activities of Lycium ferocissimum, a plant indigenous to the Cape region and Orange Free State in South Africa. The research investigates the antioxidant, cytotoxic, and anti-inflammatory properties of extracts derived from various parts of the plant, including raw fruits, ripe fruits, and leaves. Additionally, the authors conducted chemical analyses to determine the phenolic composition of these extracts.

The study is well-organized and provides a comprehensive account of the methodology employed, encompassing the extraction process, HPLC analysis, evaluation of antioxidant, cytotoxic, and anti-inflammatory activities. The findings are presented in a coherent and structured manner, supplemented with tables and figures to substantiate the results.

The authors adeptly discuss the outcomes within the context of previous research conducted on Lycium species and other related plants. Furthermore, they also well written a comprehensive conclusion that briefly summarizes the key findings and their implications for future research.

This paper is proficiently written and offers valuable insights into the potential biological activities of Lycium ferocissimum. The study contributes significantly to the existing literature on medicinal plants and their potential as sources of active compounds for drug development. After addressing the comments as suggested, I strongly recommend/ accept the article for publication.

Comments:

1. A more comprehensive clarification of the precise extraction procedure employed to derive the extracts from the plant material would greatly enhance the study.

2. It is advisable for the authors to incorporate additional details regarding the potential mechanisms that underlie the observed biological activities, particularly within the framework of the plant's traditional applications and chemical makeup.

3. To augment the statistical analysis section, it would be beneficial to expand upon the specific statistical tests employed and provide a rationale for their selection.

4. In order to amplify the paper's impact, the authors should contemplate incorporating a section that addresses the limitations of the study and outlines potential avenues for future research.

5. Investigating into the synergistic effects of the identified compounds and their potential interactions could yield valuable insights into the pharmacological potential of the plant.

6. In order to augment the investigation, it would be advantageous to contemplate carrying out mechanistic studies to elucidate the precise pathways and molecular targets implicated in the observed biological activities.

7. Additionally, broadening the scope of the study to encompass in vivo experiments could provide a more comprehensive comprehension of the plant's therapeutic potential.

In general, the research paper provides a significant contribution to the realm of natural product investigation and the study of medicinal plants.

Author Response

Dear reviewer,

File is added. 

Reviewer 5 Report

Comments and Suggestions for Authors

Comments: The submitted manuscript describes interesting bioactivity profile of an Lycium  ferocissimum miers of invasive plant. The ethanol extracts of ripe fruits and leaves of the Lycium ferocissimum species showing promising activities such as DPPH scavenging activity was 0.57 mg/mL, ABTS inhibition percentage was 88.73% at 3 mg/mL concentration, exhibited significant inhibition of viability in the Du145 cell line in the concentration range of 62.5-1000 µg/mL, and some of other LPS-induced inflammation activities. Overall, this is an interesting topic for the readers in the field and the manuscript is arranged very properly. Therefore the present version of the manuscript is should table to publish in the Plants journal.

Comments on the Quality of English Language

None

Author Response

Thank you for your valuable contribution.